# Prevalence of Post COVID-19 Condition in Primary Care: A Cross Sectional Study

**DOI:** 10.3390/ijerph19031836

**Published:** 2022-02-06

**Authors:** Patricia Montenegro, Irene Moral, Alicia Puy, Esther Cordero, Noa Chantada, Lluis Cuixart, Carlos Brotons

**Affiliations:** 1Primary Heath Care Center Sarrià, 08017 Barcelona, Spain; pmontenegro@capsarria.com; 2Teaching Unit in Family Medicine UDACEBA, 08025 Barcelona, Spain; imoral@eapsardenya.cat (I.M.); aliciapuy10@gmail.com (A.P.); ecorderofernandez@gmail.com (E.C.); noa.chantada@gmail.com (N.C.); llcuixart@eapdretaeixample.cat (L.C.); 3Research Unit, Sardenya Primary Health Care Center, 08025 Barcelona, Spain; 4Biomedical Research Institute Sant Pau, 08041 Barcelona, Spain; 5Primary Health Care Center Roger de Flor, 08013 Barcelona, Spain

**Keywords:** post-acute COVID-19 syndrome, primary health care, COVID-19, general practice, public health

## Abstract

Background: The COVID-19 pandemic is a major challenge for health systems, citizens and policymakers worldwide. It is not known how many people are affected with longer term sequelae after acute COVID-19 and a wide range of prevalence estimates have been reported with a high heterogeneity between studies. Methods: We designed a cross-sectional study to estimate the prevalence of post COVID-19 conditions in a community setting. We selected a random sample of 579 individuals from three different primary health care centers and collected information on symptoms through a standardized questionnaire. Results: Our main study finding was an overall population prevalence of 14.34% (95% CI 11.58–17.46%) of post COVID-19. Only 9% of patients were hospitalized in our study. Prevalence was higher in women than men (15.63% versus 13.06%) and the most frequent persistent symptoms were fatigue (44.6%), smell impairment (27.7%) and dyspnea (24.09%). Conclusions: The prevalence of post COVID-19 condition was lower than expected according to other studies published in the literature. The prevalence was higher in women than men, and the most frequent persistent symptoms were fatigue, smell impairment, and dyspnea.

## 1. Introduction

The COVID-19 pandemic is a major challenge for health systems, citizens and policymakers worldwide. There have been over 237 million COVID-19 cases and more than 5 million deaths worldwide, as of 19 November 2021 [1], compared with up to 650,000 deaths from influenza every year. 

At the outset of the COVID-19 pandemic, it was natural to focus first on people with severe disease who might require potentially scarce resources in hospitals and intensive care. However, we must also understand and optimize care outside of the hospital. It is time to shift the research focus to studies on living with this disease [2,3]. 

Most patients with COVID-19 make a full recovery after acute infection with SARS-CoV-2, but a significant proportion still report ongoing health problems. How many people are affected with longer term sequelae after acute COVID-19 remains unknown, but published reports indicate that approximately 10–20% of COVID-19 patients experience persistent symptoms for weeks to months following acute SARS-CoV-2 infection [4,5]. 

In the literature, the occurrence of long-term complaints from COVID-19 appears under a slew of names, including Post-Acute Sequelae of COVID-19 (PASC), Long COVID, Post-Acute COVID-19 Syndrome (PACS), Chronic COVID-19, and Long Haul COVID-19. It is commonly defined as new or persistent symptoms at 4 or more weeks from infection with SARS-CoV-2 [4].

The WHO recently leveraged a large global consensus process to publish a working clinical case definition of post COVID-19 condition, including 12 domains [5].

Post COVID-19 condition occurs in individuals with a history of probable or confirmed SARS-CoV-2 infection, usually 3 months from the onset of COVID-19 with symptoms and that last for at least 2 months and cannot be explained by an alternative diagnosis. Symptoms may be new onset following initial recovery from an acute COVID-19 episode or persist from the initial illness. Symptoms may also fluctuate or relapse over time. 

Many post-COVID-19 sufferers report a variety of these persistent and fluctuating ‘atypical’ symptoms. The most commonly reported symptoms include (but are not limited to) the following: fatigue, cough, breathlessness, fever, sore throat, chest pain, palpitations, cognitive deficits, myalgia, neurological and psychiatric symptoms, skin rashes, and diarrhea [6,7]. Estimating the prevalence of persistent symptoms is not easy due to differences in populations and methods used, and some studies reporting mixed data from hospitalized and non-hospitalized patients. Estimates therefore vary greatly across different studies [8,9,10,11,12,13,14,15,16,17].

This study aims to estimate the prevalence of Post COVID-19 condition in a community setting and see whether there are gender differences.

## 2. Materials and Methods

This prevalence study used a cross-sectional survey approach to recruit individuals to engage in the study. 

A sample size of 452 randomly selected subjects was sufficient to estimate, a population percentage considered to be around 50% with a 95% confidence interval and precision of ±5 percent units. At the time the study was designed, results around persistent COVID in primary care were still unavailable. Using a percentage of around 50% to compute sample size was a conservative approach that would give the largest sample size [18]. A replacement rate of 15% was anticipated.

We finally selected a simple random sample of 579 individuals with the following criteria (i) aged 18 to 80 years old, (ii) a confirmed SARS-CoV-2 result from a PCR, rapid antigen test or subsequent serology test performed between May and November 2020, from a total population of 92,027 individuals registered in three primary health care centers, in a community area of Barcelona, Spain. During that period, 3313 patients were diagnosed with COVID-19. Institutionalized patients, terminally ill patients, and suspected COVID-19 cases were excluded.

For study purposes, we defined post COVID-19 condition as patients with persistent symptoms for more than 4 weeks [5,7], after a COVID-19 diagnosis.

A team of trained GPs collected information on COVID-19 symptoms and length in weeks since COVID-19 diagnosis, diagnosis of pneumonia, hospitalization, ICU hospitalization, high-flow oxygen therapy, and in-hospital complications. Data were collected from all the health centers using a questionnaire, specifically developed by authors. Before its implementation, a pilot study with five patients was carried out in order to assure its applicability. 

### Statistical Analysis

Continuous variables were described by their means, SD and categorical variables by presenting their frequencies and percentages for each category. We used the chi-square test and independent Student’s *t*-test for categorical and quantitative comparisons, respectively, between men and women. STATA 14/MP was used to perform all the analyses.

The study design was reviewed and approved by the ethics committee of the Institut Universitari d’Investigació en Atenció Primària (IDIAP Jordi Gol i Gurina) (number 20/140-PCV). 

## 3. Results

Medical records from 579 patients from three primary health centers were reviewed between April and June 2021. The mean age was 45.8 years old (SD 16.2) and 50.3% were males. The mean number of days from onset of symptoms to the interview was 191.97 days (27.42 weeks). The characteristics of the study population are summarized in Table 1. 

During the acute phase of the disease, a total of 64 (11%) patients were diagnosed with pneumonia and 54 (9.3%) were hospitalized (mean length of stay was 13.48 days), 35 (6%) were under high-flow oxygen therapy, and 7 (1.2%) required ICU management. 

The most frequent acute symptoms were cough (48%), fever (44%), fatigue (33.7%), headache (28.3%), smell impairment (26.4%), taste impairment (20.4%) and myalgia (18.6%). The list of symptoms recorded in the medical records is shown in Table 2. 

At the time of the evaluation, 83 (14.3%, 95% CI 11.6–17.5%) patients had persistent symptoms, 47% having at least two different symptoms. The presence of PASC was higher in women (*n* = 45, 15.6%) than men (*n* = 38, 13.1%), although there were no statistically significant differences. Women with PASC were on average 4 years older than men with PASC. A wide range of persistent symptoms was present during the acute phase and at 4 weeks after the COVID-19 diagnosis (Table 3). There were symptoms that had disappeared at 4 weeks such as wheezing, runny nose, sickness/vomiting, and confusion and symptoms that persisted only in men (fever and abdominal pain) or only in women (shaking chills, nasal congestion, anorexia, and diarrhea).

The most frequent persistent symptoms among the 83 patients were fatigue (44.6%) with a mean length of 17.35 weeks, smell impairment (27.7%) with a mean length of 17.68 weeks, dyspnea (24.1%) with a mean length of 17.9 weeks, taste impairment (18.1%) with a mean length of 19.86 weeks, cough (15.6%) with a mean length of 17.4 weeks, headache (13.2%), with a mean length of 16.7 weeks, and chest pain (9.6%) with a mean length of 20.8 weeks.

The most frequent persistent symptoms in men with post COVID-19 condition were fatigue (39.5%), dyspnea (31.6%) and anosmia (21.1%), while in women they were fatigue (48.9%) and anosmia (33.3%) (Figure 1). 

We also found that 4% of patients experienced a worsening of a previous condition, and 3.1% of patients had suspected permanent sequelae related to their COVID-19 infection.

## 4. Discussion

The main study finding was an overall population prevalence of 14.34% of post COVID-19 condition in a community setting. The most frequent symptoms among patients with post COVID-19 condition were fatigue (44.6%), smell impairment (27.7%) and dyspnea (24.1%). 

Studies conducted in different regions of the world showed a wide range of results. To illustrate the geographic heterogeneity seen in post COVID-19 condition prevalence estimates, specific studies from Germany, USA, Italy, and China reported prevalences of 28% 35%, 51% and 76%, respectively [19,20].

Three systematic reviews showing long COVID results have been published in the literature. Lopez-Leon et al. [21] analyzed data from 15 studies (preprinted article), estimating that 80% (95%CI 65–92%) of patients infected with SARS-CoV-2 developed one or more long-term symptoms. The five most common symptoms were fatigue (58%), headache (44%), attention disorder (27%), hair loss (25%) and dyspnea (24%). However, the studies were very heterogeneous, mainly due to the follow-up time references and mixture of patients who had moderate and severe COVID-19. In addition, several studies that used a self-reported questionnaire could have resulted in reporting bias. 

Chen et al. conducted a meta-analysis of 29 studies published in the literature (preprint article), reporting a pooled prevalence of post COVID condition of 43% (95%CI 35–63%) [22].

However, the authors acknowledged a substantial heterogeneity among the included studies, with estimates ranging from 9% to 81%. Each study differed in the methods employed and the COVID-19 study population, making the results not entirely comparable. The authors found a pooled PASC prevalence in hospitalized patients of 57% (95%CI 45–68%) compared to the estimate in a mix of hospitalized and non-hospitalized COVID-19 patients of 31% (95%CI 24–40%). Nevertheless, it was noted that a wide range of estimates still contributed to both groups.

Groff et al. [23] published a systematic review of 57 studies with 250,351 COVID-19 survivors. They found a median proportion of COVID-19 survivors experiencing at least 1 PASC of 54.0% (95%CI 45.0–69.0%; 13 studies) at 1 month (short-term), 55.0% (95%CI 34.8–65.5%; 38 studies) at 2 to 5 months (intermediate-term), and 54.0% (95%CI 31.0–67.0%; 9 studies) at 6 or more months (long-term). The most prevalent pulmonary sequelae, neurologic disorders, mental health disorders, functional mobility impairments, and general and constitutional symptoms were chest imaging abnormalities (62.2%, 95%CI 45.8–76.5%), difficulty concentrating (23.8%, 95%CI 20.4–25.9%), generalized anxiety disorder (29.6%, 95%CI 14.0–44.0%), general functional impairments (44.0%, 95%CI 23.4–62.6%), and fatigue or muscle weakness (37.5%, 95%CI 25.4–54.5%), respectively.

However, the authors were unable to stratify the risk of PASC by severity of initial illness or by preexisting comorbidities, patient age, or other factors that may affect an individual patient’s risk of PASC. 

Our post COVID-19 condition estimate of 14.34% is considerably lower than the 43%, 54% and 80% figures provided by Chen [22], Groff [23] and Lopez-Leon [21], respectively. One of the reasons for this difference is that in our study only 9% of patients were hospitalized, while in the systematic reviews most of the studies were done on hospitalized or mixed hospitalized/non-hospitalized patients. Individuals who were hospitalized during acute COVID-19 infection had a higher PASC prevalence than non-hospitalized patients. 

The most prevalent sequelae described in the two systematic reviews was fatigue at 58% [22] and 23% [23], respectively. This result is in keeping with the one observed in our study, where fatigue was also the most prevalent persistent symptom. 

Women had a higher prevalence of post COVID-19 condition PASC than men (15.63% vs. 13.06%), although the result was not statistically significantly different. Groff et al. [23] estimated a pooled PASC prevalence in women of 49% higher than that of men, with 37%. 

One limitation of our study is that we obtained the medical records of COVID-19 patients from three primary health care centers for data review. Symptoms could have been under-reported either because the patients did not complain about them or because the doctors did not accurately describe them. We therefore believe this might have introduced a bias, and the prevalence of post COVID condition might be underestimated. 

Despite such limitations, we are reasonably confident of the generalizability of our results, since the sample of participants was randomly selected from the overall population diagnosed with COVID-19 in three different primary health care centres, and represented an urban community in the primary care setting in Spain. A further strength is that staff were specifically trained to conduct the study in the community. 

## 5. Conclusions

The prevalence of post COVID-19 condition in a community setting was lower than expected according to other studies published in the literature. The prevalence was higher in women than men, and the most frequent persistent symptoms were fatigue, smell impairment and dyspnea. 

## Figures and Tables

**Figure 1 ijerph-19-01836-f001:**
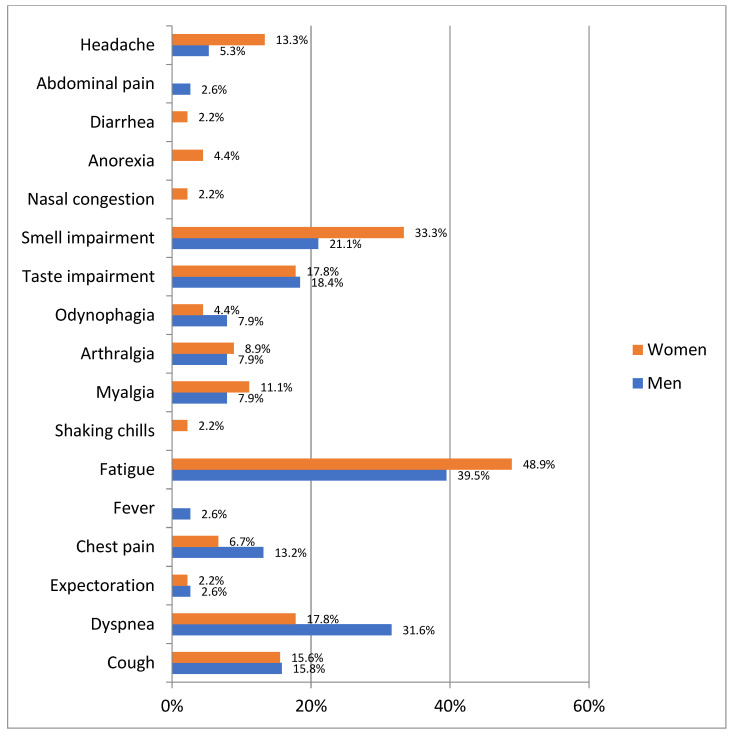
Prevalence of persistent symptoms among patients with post COVID-19 condition by gender (*n* = 83).

**Table 1 ijerph-19-01836-t001:** Characteristic of the study population.

	Men (*n* = 291)	Women (*n* = 288)	All (*n* = 579)
Age, mean (SD)	45.19 (16.51)	46.46 (15.92)	45.82 (16.22)
Smokers, *n* (%)	35 (12.03%)	25 (8.68%)	60 (10.36%)
Hypertension, *n* (%)	37 (12.71%)	35 (12.15%)	72 (12.44%)
Dyslipidemia, *n* (%)	46 (15.81%)	42 (14.58%)	88 (15.20%)
Diabetes, *n* (%)	18 (6.19%)	8 (2.78%)	26 (4.49%)
Obesity, *n* (%)	31 (10.65%)	30 (10.42%)	61 (10.54%)
Cardiac condition, *n* (%)	13 (4.47%)	4 (1.39%)	17 (2.94%)
Psychiatric condition, *n* (%)	34 (11.68%)	36 (12.50%)	70 (12.09%)
Autoimmune disease, *n* (%)	4 (1.37%)	11 (3.82%)	15 (2.59%)
COPD, *n* (%)	6 (2.06%)	10 (3.47%)	16 (2.76%)
Asthma, *n* (%)	16 (5.50%)	15 (5.21%)	31 (5.35%)
Cerebrovascular disease, *n* (%)	6 (2.06%)	4 (1.39%)	10 (1.73%)
Chronic renal disease, *n* (%)	2 (0.69%)	4 (1.39%)	6 (1.04%)

**Table 2 ijerph-19-01836-t002:** List of most common symptoms in the acute phase of COVID-19.

	Men (*n* = 291)	Women (*n* = 288)	All (*n* = 579)
Cough, *n* (%)	141 (48.45%)	137 (47.57%)	278 (48.01%)
Dyspnea, *n* (%)	45 (15.46%)	42 (14.58%)	87 (15.03%)
Expectoration, *n* (%)	11 (3.78%)	5 (1.74%)	16 (2.76%)
Chest pain, *n* (%)	18 (6.19%)	26 (9.03%)	44 (7.60%)
Fever, *n* (%)	146 (50.17%)	109 (37.85%)	255 (44.04%)
Fatigue, *n* (%)	88 (30.24%)	107 (37.15%)	195 (33.68%)
Shaking chills, *n* (%)	11 (3.78%)	8 (2.78%)	19 (3.28%)
Wheezing, *n* (%)	3 (1.03%)	5 (1.74%)	8 (1.38%)
Myalgia, *n* (%)	53 (18.21%)	55 (19.10%)	108 (18.65%)
Arthralgia, *n* (%)	49 (16.86%)	46 (15.97%)	95 (16.41%)
Odynophagia, *n* (%)	42 (14.43%)	49 (17.01%)	91 (15.72%)
Taste impairment, *n* (%)	56 (19.24%)	62 (21.53%)	118 (20.38%)
Smell impairment, *n* (%)	78 (26.80%)	75 (26.04%)	153 (26.42%)
Runny nose, *n* (%)	30 (10.31%)	28 (9.72%)	58 (10.02%)
Nasal congestion, *n* (%)	31 (10.65%)	30 (10.42%)	61 (10.54%)
Anorexia, *n* (%)	4 (1.37%)	8 (2.78%)	12 (2.07%)
Diarrhea, *n* (%)	31 (10.65%)	28 (9.72%)	59 (10.19%)
Sickness/vomiting, *n* (%)	9 (3.09%)	13 (4.51%)	22 (3.80%)
Abdominal pain, *n* (%)	4 (1.37%)	12 (4.17%)	16 (2.76%)
Confusion, *n* (%)	4 (1.37%)	2 (0.69%)	6 (1.04%)
Headache, *n* (%)	80 (27.49%)	84 (29.17%)	164 (28.32%)

**Table 3 ijerph-19-01836-t003:** Prevalence of persistent symptoms among people with the symptom during the acute phase.

	Men	Women	All
Cough, *n* (%) [N]	6 (4.26%) [141]	7 (5.11%) [137]	13 (4.68%) [278]
Dyspnea, *n* (%) [N]	12 (26.67%) [45]	8 (19.05%) [42]	20 (22.99%) [87]
Expectoration, *n* (%) [N]	1 (9.09%) [11]	1 (20.00%) [5]	2 (12.50%) [16]
Chest pain, *n* (%) [N]	5 (27.78%) [18]	3 (11.54%) [26]	8 (18.18%) [44]
Fever, *n* (%) [N]	1 (0.68%) [146]		1 (0.39%) [255]
Fatigue, *n* (%) [N]	15 (17.05%) [88]	21 (19.63%) [107]	36 (18.46%) [195]
Shaking chills, *n* (%) [N]		1 (12.50%) [8]	1 (5.26%) [19]
Myalgia, *n* (%) [N]	3 (5.66%) [53]	5 (9.09%) [55]	8 (7.41%) [108]
Arthralgia, *n* (%) [N]	3 (6.12%) [49]	4 (8.70%) [46]	7 (7.37%) [95]
Odynophagia, *n* (%) [N]	2 (4.76%) [42]	2 (4.08%) [49]	4 (4.40%) [91]
Taste impairment, *n* (%) [N]	7 (12.50%) [56]	8 (12.90%) [62]	15 (12.71%) [118]
Smell impairment, *n* (%) [N]	8 (10.26%) [78]	14 (18.67%) [75]	22 (14.38%) [153]
Nasal congestion, *n* (%) [N]		1 (3.33%) [30]	1 (1.64%) [61]
Anorexia, *n* (%) [N]		2 (25.00%) [8]	2 (16.67%) [12]
Diarrhea, *n* (%) [N]		1 (3.57%) [28]	1 (1.69%) [59]
Abdominal pain, *n* (%) [N]	1 (25.00%) [4]		1 (6.25%) [16]
Headache, *n* (%) [N]	4 (5.00%) [80]	7 (8.33%) [84]	11 (6.71%) [164]

*n*: number of people with the persistent symptom; [N]: number of people with the symptom during the acute phase.

## Data Availability

The data presented in this study is available on request from the corresponding author.

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
