# Peer review of "Prevalence of Post COVID-19 Condition in Primary Care: A Cross Sectional Study"

_ijerph, 2022, doi:10.3390/ijerph19031836_

Round 1
Reviewer 1 Report
The paper by Montenegro Lafont et al estimates the prevalence of post covid-19 conditions in a community setting. It concludes that the prevalence of post COVID-19 condition was lower than expected according to other studies published in the literature. Even though there are several studies reporting the prevalence of post covid-19 conditions, most of them have been developed in a hospital setting. Thus, this population based study is relevant. However, there are some major issues to be improved and clarified. I detailed them below.
Introduction
Lines 36-37: Must be rewritten. Incomprehensible phrase.
Lines 62-63: The study aim must be justified. The authors should clearly state why it was necessary to conduct this study.
Materials and Methods
Lines 67-69: Parameters included in sample size calculation must be justified. Why did authors consider 50% population percentage of post Covid-19 conditions? Is this value reported in previous studies? How the authors decided that the required precision was 5%? What does the replacement rate mean?
Line 70: Please provide more details about the randomisation: method used to generate the random allocation sequence, who generated the allocation sequence, etc.
A subsection must be added detailing all methods used for statistical analysis.
Results
I would recommend using only one decimal to report percentages.
Line 86: According to the material and methods section, patients included had a SARS-CoV-2 confirmation test between May and November 2020. However, according to the results section, medical records from 579 patients were reviewed between April to November 2021. Please clarify.
Tables 1, 2 and 3: Please add a column to describe the characteristics, most common symptoms and persistent symptoms of entire sample. Why have men and women patients been compared in these tables? Introduction section does not mention any aim related to gender differences.
Lines 91-96: Please use the following format “n (%)”in case you are providing percentages not reported in tables. Otherwise, absolute frequencies are unknown. Consider this point for all results.
Line 101: I guess you mean “statistically significant differences”.
Lines 101-102: Exact values must be reported in the main text in case these are not provided in tables.
Lines 102-103: What does a persistent symptom mean? To my understanding, a symptom is persistent if it is present during the acute phase and at 4 weeks. If this is true, the following phrase doesn’t make sense: “A wide range of persistent symptoms was present 4 weeks after the diagnose of 102 COVID-19 (Table3)”. According to the paper, table 3 represents prevalence of persistent symptoms. Results conclude that there is no patient with runny nose at 4 weeks. Is runny nose a persistent symptom? All these must be clarified. I recommend authors to define clearly what a persistent symptom is.
Table 3: Results reported as “n(%)[N]” are not clear. Instead I recommend create new columns for “N” (number of patients with symptoms during the acute phase) to be added in table 3.
Lines 108-109: Figure 3 doesn’t exist. I guess you mean Figure 1. According to the text it is representing the most frequent persistent symptoms among the 83 patients with persistent symptoms. But according to the figure title it is prevalence of symptoms among people with persistent symptoms by gender. Is it the same “a symptom” and “a persistent symptom”? As I previously recommended, please define clearly what a persistent symptom is. Moreover, percentages represented in figure 1 are not reported in the paper. Only few overall percentages were provided in the text (from line 109 to 113). Thus, it is necessary to create a new table or add new columns in table 3 to report percentages represented in figure 1.
Lines 109-113: Was the follow-up period the same for all patients to evaluate persistent symptoms? If it was not, authors must report mean follow-up period and its range. I guess some symptoms were present beyond four weeks. Which value is imputed to compute duration of symptom if you have a patient with a symptom present until the end of follow-up? Did authors consider censoring?
Lines 116-119: What do these values refer? For example, it says the most frequent PASC in men were fatigue (39.47%). This percentage (39.47%) doesn’t correspond to figure 1 (the bar of fatigue in men was smaller than 20%) neither table 3 (fatigue in men was 17.05%). What do they represent? It is necessary to create a new table or add new columns in table 3 to report these percentages.
Discussion
Lines 125-126: Probably it is much more accurate to say “The most frequent symptoms among people with persistent symptoms were fatigue (44.6%), smell impairment (27.7%) and dyspnea (24.09%)”
Line 147: It is unclear what the following text means: “revealing a sizeable difference patient of 57% (95%CI 45%-68%)”. Please clarify.
Lines 161-163: It is unclear what the following text means “However, authors were not able to stratify the risk of PASC by severity of initial 161 illness (for example, community-based vs hospitalized vs required care in an intensive 162 care unit vs required invasive life-sustaining measures)”. Please clarify.
Lines 172:174: This sentence must be reviewed. It is unclear.
Line 176: Instead “statistically different” use “statistically significant difference”.
Line 185: Please detail and justify the generalizability of your results and conclusions.
Line 188: Instead of “personnel” use “staff”.
Lines 187-189: I disagree. This cannot be considered as a strength of a scientific study. All scientific studies must follow a specific protocol and in case it is required, some staff are trained to conduct the study. This is something that all we expect.
References
Number of references is limited. Taking into account high number of research papers added in Covid-19 literature every day, I highly recommend to authors update their literature search and extend the number of references.
Additionally to my previous comments, the paper requires proofing by a native speaker.
Reviewer 2 Report
This is a cross-sectional study from symptom reporting gathered from medical records and the limitations of the study are discussed. It might be worthwhile to define in more detail the methodology as to how the symptom reporting data was collected and documented and what is the nature of the standardized questionnaire used for data collection. This can be expanded as to how the questionnaire was standardized and validated. What was the nature of the training of the GP's who collected the data? The Materials and Methods section needs more detail. Good manuscript and it is well conceived study given the limitations of cross-sectional data from a marginally adequate sample size.
Reviewer 3 Report
- Same number of decimal points should be maintained throughout the manuscript like either two or three.
- Value labels can be added in Figure-1 and bars can be rearranged chronologically.
- In discussion part, the author mentioned “lack of power of the study” in line 176-177. The author need to describe or justify this concept broadly in method section – how this come?
- Instead of sharing numbers in conclusion, it is better to rewrite conclusion in a general way.
- Literature review need to be more enriched and thus reference list.
- Further analysis is recommended (if data available) to see differences in prevalence of different symptoms among different age groups.
Round 2
Reviewer 1 Report
All my queries have been answered. No more comments.